# Pulmonary Vascular Remodeling in Pulmonary Hypertension

**DOI:** 10.3390/jpm13020366

**Published:** 2023-02-19

**Authors:** Zhuangzhuang Jia, Shuai Wang, Haifeng Yan, Yawen Cao, Xuan Zhang, Lin Wang, Zeyu Zhang, Shanshan Lin, Xianliang Wang, Jingyuan Mao

**Affiliations:** 1Department of Cardiovascular Diseases, First Teaching Hospital of Tianjin University of Traditional Chinese Medicine, National Clinical Research Center for Chinese Medicine Acupuncture and Moxibustion, Tianjin 300381, China; 2Tianjin University of Traditional Chinese Medicine, Tianjin 301617, China; 3Department of Cardiovascular Diseases, The First Affiliated Hospital of Henan University of Traditional Chinese Medicine, Zhengzhou 450000, China

**Keywords:** pulmonary hypertension, pulmonary vascular remodeling, intima, media, adventitia, extracellular matrix

## Abstract

Pulmonary vascular remodeling is the critical structural alteration and pathological feature in pulmonary hypertension (PH) and involves changes in the intima, media and adventitia. Pulmonary vascular remodeling consists of the proliferation and phenotypic transformation of pulmonary artery endothelial cells (PAECs) and pulmonary artery smooth muscle cells (PASMCs) of the middle membranous pulmonary artery, as well as complex interactions involving external layer pulmonary artery fibroblasts (PAFs) and extracellular matrix (ECM). Inflammatory mechanisms, apoptosis and other factors in the vascular wall are influenced by different mechanisms that likely act in concert to drive disease progression. This article reviews these pathological changes and highlights some pathogenetic mechanisms involved in the remodeling process.

## 1. Introduction

Pulmonary hypertension (PH) is a life-threatening disorder characterized by elevated pressure in the pulmonary arteries due to increased pulmonary vascular resistance [1]. PH is currently defined by a mean pulmonary artery pressure (mPAP) greater than 20 mm Hg on supine right heart catheterization at rest [2]. This definition differs from the previous threshold of 25 mm Hg or greater in recognition that patients with mPAP of 21 mm Hg to 24 mm Hg are at increased risk of mortality and hospitalization compared with those with an mPAP of 20 mm Hg or lower [3]. As a severe cardiopulmonary disease, PH is characterized by vascular remodeling and occlusion of small pre-capillary pulmonary arteries. Alterations in pulmonary vascular structure and function can lead to resistance to blood flow and the possible development of right-sided heart failure, leading to severe morbidity and mortality. There are several types of PH, which can be familial or secondary to an underlying disease [4]. Regardless of the etiology, the exact pathophysiological mechanisms leading to PH development and progression are primarily unidentified [5].

PH is a panvasculopathy, meaning all layers of the vascular wall are involved, which is also reflective of gene-environment interactions and has essential genetic and epigenetic mechanisms. The mechanism of PH is complex, and pulmonary vascular remodeling is the key pathological feature of PH. Pulmonary vascular remodeling involves the proliferation and phenotypic transformation of pulmonary artery endothelial cells (PAECs) and pulmonary artery smooth muscle cells (PASMCs) of the middle membranous pulmonary artery, as well as complex interactions involving external layer pulmonary artery fibroblasts (PAFs) and extracellular matrix (ECM). Inflammatory mechanisms, apoptosis and other factors in the vascular wall are influenced by different mechanisms that likely act in concert to drive disease progression (Figure 1) [6]. Here, we highlight fundamental changes in the pathobiology of PH vascular remodeling and discuss potential therapeutic interventions that may prevent disease progression and even reverse PH vascular remodeling.

## 2. Intima Remodeling in Pulmonary Vascular Remodeling

The intima represents the thick interface of endothelial cells between the media and flowing blood. The endothelial cells provide a broad unobstructed flow surface area, contributing to the suitable perfusion pressure that is the normal state of pulmonary circulation [7]. Patients with severe PH have an approximately 3 fold increase in pulmonary intima, but the mechanism of intima damage is unclear [8] and intimal thickening will result in an approximately 40-fold increase in resistance of the pulmonary vascular. There are various types of intimal thickening and PAECs dysfunction, as the most critical component of the intima accelerates the process of intima remodeling [9]. Under normal physiological conditions, PAECs are in a steady state and secrete a variety of active factors that disturb PAECs and PASMCs proliferation, coagulation, the attraction of inflammatory factors and activation of vasoactivity, which leads to dysfunction and pathological changes of PASMCs in PH [10]. Multiple factors trigger PAECs dysfunction in PH [11,12,13,14], such as shear stress, hypoxia, inflammation, PAECs phenotypes, the bone morphogenic type 2 receptor (BMPR2), and cilia length [15,16,17] (Figure 2). Meanwhile, the deterioration of endothelial metabolic function in the pulmonary vascular system is becoming an important driver of PAECs dysfunction and PH development [18].

### 2.1. Phenotypes of PAECs Dysfunction in Intima Remodeling

Damage and apoptosis of PAECs can occur in the early stages of PH pathogenesis, while anti-apoptotic PAECs appear later as PH progresses [19]. In late PH, hyperproliferative and anti-apoptotic PAECs predominate and facilitate the formation of plexiform lesions [20]. The pathogenesis of PH is usually associated with abnormal endothelial cell barrier integrity, and patients with idiopathic PH (iPH) often exhibit a hypercoagulable phenotype. Additionally, there is a growing awareness that complex alterations in metabolic and epigenetic pathways facilitate the progression of PH [14]. However, it is essential to note here that PAECs include separate subpopulations of endothelial cells, which are possible exposure to multiple adverse stimuli and physical damages depending on location in the pulmonary vascular system [21].

Endothelial-mesenchymal transition (EndoMT) is a phenotypic change in which PAECs manifest a mesenchymal-like phenotype with concomitant endothelial cell characteristics loss while upregulating the level of mesenchymal markers. Furthermore, PAECs adopt highly migratory and invasive cell phenotype characteristics with loss of cell-cell contact [22]. Strongly expressed α-smooth muscle actin (α-SMA), vimentin and VE-cadherin appeared in the PAECs of human PH patients and PH rat models induced by monocrotaline (MCT) hypoxia accompanied EndoMT. PAECs treated with transforming growth factor beta (TGF-β) induce the levels of the EndoMT transcription factors TWIST1, SNAIL1 and the previously mentioned mesenchymal markers involved in this process [23,24]. More interestingly, BMP-7 was abrogated in hypoxia-induced PAECs by the action of EndoMT via inhibiting the mTORC1 signaling pathway [25]. Low BMPR2 expression favors EndoMT leading to over-activated TGF-β signaling [26]. In conclusion, altered TGF-β/BMP signaling is associated with the EndoMT process in PH [27]. Hypoxia acts as an inducer of EndoMT via increasing hypoxia-inducible transcription factor-1α (HIF-1α) and hypoxia-inducible transcription factor-2α (HIF-2α) in PH [28]. Finally, microRNA, such as miR-27a, miR-124 and miR-181b, can be implicated in EndoMT in PH [29,30,31] (Figure 3).

### 2.2. PAECs Survival and Proliferation in Intima Remodeling

BMP receptor signaling, which is encoded by SMAD1, SMAD4 and SMAD9, plays an important role in PH development. BMPR2, as a transmembrane enzyme receptor that regulates TGF-β signaling in PAECs in the lumen of the pulmonary vessels, promotes the survival of PAECs and antagonizes PASMCs proliferation [32,33]. Interestingly, GDF2 encodes the circulating BMP 9, which is a ligand for the BMP2 receptor, and mutations in GDF2 reduced levels of BMP family expression [34].

Additionally, siRNA-mediated silencing of BMPR2 in PAECs contributes to the inhibition of Ras/Raf/ERK and Ras signaling reversing proliferation and hypermotility [35]. With the further development of pathology in PH, PAECs proliferation is a major manifestation resulting in complex arterial structural and functional remodeling, and multiple pathways regulate this transition. Peroxisome proliferator-activated receptor-γ in PAECs inhibits the cell cycle and disrupts endothelial cell barrier function, while antagonizing the migration and angiogenic properties of PAECs [36]. Furthermore, recent studies have suggested a role for endothelial prolyl hydroxylase 2 (PHD2) in PH pathology, and mice with Tie2Core-mediated PHD2 disruption in PAECs exhibited vascular remodeling in PH [37].

The proliferation and survival of PAECs are influenced by several other factors that also exacerbate the pathological condition of PH, such as disruption of Cav1 [38]. mTOR, Nur77 and GDF11 also act as inhibitors of PAECs proliferation and angiogenesis after hypoxia [39,40]. Oxidative, antioxidant and nitrification equally affect endothelial function. Inhibition of reactive oxygen species (ROS)-induced Ca^2+^ entry also downregulates the migration and proliferation of PAECs [41]. It has been recently shown that endostatin, a cleavage product of Col18A1, inhibits PAECs proliferation and apoptosis via CD47 and ID1/TSP-1/CD36 signaling [42]. The absence of Notch coupling to Sox17 in endothelial cells may exacerbate PH by upregulating the monolayer vulnerability of PAECs [43]. These findings demonstrate the complex relationship between PAECs survival and proliferation in PH.

### 2.3. PAECs Activation and Thrombogenicity in Intima Remodeling

The presence of thrombotic lesions is a common pathological manifestation of PH. However, the role assumed by thrombus in PH remains controversial [44]. Multiple factors participate in the regulation of PH by affecting PAECs activation and thrombogenicity. A few studies have shown that coagulation factors, represented by activators of the coagulation cascade, lead to the aggregation of fibrin clots and blockage of blood vessels and exacerbate PAECs dysfunction, leading to vascular remodeling [45]. The levels of von Willebrand Factor (vWF) in PH patients also increase in the plasma. PAECs and platelets express and release vWF when activated, facilitating their interaction. The levels of thrombomodulin, as a series of anti-coagulant factors, are decreased in PH patients, which can inhibit this deterioration by ingesting prostacyclin, vasodilators or tadalafil [46]. CD40L is an inflammatory factor that cleaves into sCD40L upon activation, which is known to promote significantly in PH patients, eventually contributing to vascular remodeling in PH [47]. Although substantial evidence suggests that platelets and thrombogenicity exacerbate the pathogenesis of PAECs dysfunction, the molecular mechanisms need further elucidation.

### 2.4. PAECs Metabolism and Epigenetics in Intima Remodeling

Factors affecting PAECs metabolism and epigenetics participate in the regulation of PH. Metabolic abnormalities, particularly aerobic glycolysis or the Warburg effect, have been proposed as important pathogenic mechanisms in developing PH. PFKFB3 is an essential regulator of glycolysis, and its deficiency inhibits pulmonary vascular remodeling [48]. Endothelin 1/eNOS signaling also serves as an essential pathway that regulates the glycolytic process [49]. Further studies suggest that BolA family member 3 (BOLA3) is involved in the operation of glycolysis and mitochondrial respiratory function [50]. Epigenetic mechanisms are also generally considered to be important in regulating PAECs metabolism. The delivery of glutamine carbon into the tricarboxylic acid (TCA) cycle becomes active in PAECs in the case of mutations in the BMPR2 gene, and the strict requirement for glutamine is driven by the loss of deacetylase sirtuin 3 activities. Additionally, the pharmacological effects of glutaminase can be targeted to reduce the severity of PH pathology [51]. The distribution of other genetic variants showed that variants in ACVRL1, ENG, SMAD9, KCNK3 and TBX4 contributed to PH [52], but only about 1% of the cases in each gene. Therefore, these factors are emerging as promising targets for PH treatment.

## 3. Media Remodeling in Pulmonary Vascular Remodeling

The media, composed mainly of PASMCs, is the focus of PH because of its property mediating hypoxic pulmonary vasoconstriction. No single property can account for the complexity of the contribution of PASMCs to pulmonary vascular remodeling [53].

### 3.1. Heterogeneity of PASMCs Phenotypes and Functions

Most PASMCs found during embryogenesis or mesothelium via EndoMT originate from the mesoderm [54,55,56]. Heterogeneity and phenotypic plasticity of different subgroups of PASMCs act as pivotal parts in regulating vascular functions and adapting to changes in microenvironmental cues of pulmonary circulation [57,58]. To acquire or lose proliferative and migratory potential and synthesize ECM, PASMCs can be reversibly converted from resting to contractile or synthetic in response to specific stimuli [59]. Multiple growth factors and inflammatory mediators, such as TGF-β, platelet-derived growth factor (PDGF), angiotensin-II; as well as stimuli, such as mechanical forces, epigenetics and ECM tissue heterogeneity, are essential regulators facilitating PASMCs transformed from the contractile to the synthetic phenotype [60,61,62]. PASMCs acquire migratory and proliferative capacities during this shift to achieve remodeling of the media membrane. There are several PASMCs-specific genes that can be used to determine differentiation status. Most extensively studied α-SMA, smooth muscle myosin heavy chain, and smoothelin, but smooth muscle 22a (SM22α), meta-vinculin, desmin and smooth muscle calponin also characterize the mature contractile PASMCs phenotype [63,64,65]. The majority of such proteins can be used as contractile machinery components and regulators of contraction, whose level expression is downregulated during the switch to the dedifferentiated/synthetic phenotype. Other markers, such as collagen type I, SMemb/non-muscle myosin heavy chain isoform-B, connexin-43, cellular retinol-binding protein-1, osteopontin (OPN) and syndecan-1/-4, are expressed during transforming to the synthetic phenotype [66,67,68]. The phenotypic variability of PASMCs is also regulated by various microRNAs, such as miR-125a-5b, miR-133, and miR-193b [69,70,71].

### 3.2. Inherent Intrinsic Abnormalities in PASMCs

In the process of PH, pulmonary vascular remodeling involves alterations in the phenotype of the various constricted vascular cells that make up the arterial wall, including PASMCs, PAFs and myofibroblast-like cells [72,73,74]. There is growing evidence that some different types of cells, such as PASMCs, adventitial fibroblasts and myofibroblast-like characteristics cells, exhibit a range of inherent intrinsic abnormalities. The intrinsic abnormalities, including the dysfunction of BMPR2 signaling, dysregulation of K^+^ channel homeostasis, inhibition of peroxisome proliferator-activated receptor-γ and forkhead box O-1 (FOXO-1) signaling, stimulating activation of the nuclear factor-kappa B (NF-κB) and PHD2/HIF-1α signaling pathway, contribute to intensification of proliferation and phenotype transformation [75,76,77]. The nuclear factor of activated T cells (NFAT) is activated in PASMCs within pulmonary circulation in both animal and human PH suggesting that NFAT plays a critical role in the pathogenesis of PH. NFATc1, NFATc2 and NFATc3 are the main isotypes of NFAT, which is characterized by inflammation, hyperproliferation and metabolic change [78].

Diminished heterozygous function of BMPR2 and several other genes involved in PH, such as activin receptors like kinase 1(ALK1), ALK6 (also known as BMPR1B), endoglin (ENG), and BMP9, have promoted several pathological manifestations of PH, such as depolarization, anti-apoptosis, contraction and proliferation of PASMCs [79,80,81]. In experimental models, the above changes were also shown to be associated with alteration of the ion channel, such as inhibiting RhoA/Rho signaling reduced Ca^2+^ sensitization and vascular remodeling and lowering pulmonary pressure. Furthermore, a heightened response to PDGF, epidermal growth factor (EGF) and fibroblast growth factor-2 (FGF-2) was also confirmed in the contractile cell type of PH [82,83,84]. It is precisely because the understanding of potential mechanisms is still limited that a better exploration of the inherent intrinsic abnormalities in PASMCs is needed.

### 3.3. Energy and Metabolic Changes in Media Remodeling

Vasoconstrictor cells from PH patients can adjust their energy and metabolic state to meet changing bioenergetic and biosynthetic demands [85]. It has been demonstrated that the vascular remodeling process of PH involves glycolytic processes regulated by mitochondrial oxidative phosphorylation and glycolysis, dysregulation of amino acid metabolism and lipid oxidation pathways; and NFATc2 plays an important regulatory role in these processes [86]. Furthermore, promoting oxidative metabolism via attenuating pyruvate dehydrogenase kinase (PDK) and lacking malonyl-CoA decarboxylase can reverse pulmonary vascular remodeling in PH mice. PASMCs in iPH also show other pathological metabolic pathways such as activation of hypoxia-inducible factor, enhanced glutamine metabolism and activation of fatty acid oxidation [87,88]. It is evident that energy and metabolic changes have essential effects on vascular remodeling in PH.

### 3.4. Altered Cell Communications and Senescence in Media Remodeling

Interactions between pulmonary vascular cells, such as PAECs, PASMCs, PAFs, pericytes and immune cells, are the basis for the structure and function of the blood vessels in the lung, including stability of related cellular phenotypes, intrinsic properties and their responsiveness to external microenvironmental stimuli [89]. Alterations of PASMCs communicate with PAECs and play a major regulatory role in the pathological development of PH [90]. PAECs in PH lose the ability to modulate the appropriate moderate balance between vasoconstrictors, such as endothelin-1 (ET-1) and serotonin, and vasodilators, such as nitric oxide (NO) and prostacyclin [91,92,93,94]. The capacity of PAECs is to maintain a regular vascular network by secreting vasoactive factors, which affects the interactions between PAECs and PASMCs, and regulates multicellular contraction, proliferation, and survival. In particular, inflammation often precedes vascular remodeling, which strongly supports the importance of multicellular intercommunication.

Senescent vascular PAECs are characterized by morphologic and metabolic changes, and the acquisition of a senescence-associated secretory phenotype (SASP) [95]. Furthermore, senescent vascular smooth muscle cells have emerged as key triggers of vascular structural disruption and remodeling in PH [96]. OPN, p53 and p16Ink4a signaling are thought to play central roles in the cellular senescence program of PH [97,98]. Moreover, the SASP of PASMCs exacerbates pathological overproliferation of cells and promotes critical media remodeling pathways [99]. How well the properties of senescent PASMCs are understood is critical because cellular senescence leads to irreversible PH-associated pulmonary vascular remodeling.

## 4. Adventitia Remodeling in Pulmonary Vascular Remodeling

Until recently, the adventitia has somehow been overlooked in this traditional concept, but emerging evidence leads us to recognize the possibility that the adventitia can serve as a staging ground for some of the earliest changes that occur in the vessel wall, especially inflammatory changes [100]. Cumulative results have suggested that the adventitia is not merely a bystander in the pathogenesis of arterial disease but may represent a direct driving force for the development of PH. The adventitia consists mainly of a connective tissue sheath surrounding the pulmonary arteries and PAFs are crucial components [101]. During the period of vascular remodeling induced by PH, PAFs are highly activated and undergo phenotypic transformation characterized by hyperproliferation, migration and inflammatory activities (Figure 4).

### 4.1. Hypoxia Induces Changes in Adventitia Structure

Hypoxia is the pathological basis of pulmonary vascular remodeling. The thickening of adventitia led to decreased vascular compliance and distensibility and increased blood pressure and right ventricular dysfunction. The adventitia structure change is considered the earliest and most prominent structural change in the PH process [102]. The change of adventitia is earlier than that of intima and media, which can induce the degeneration and thickening of media. Hypoxia causes the thickness of adventitia to increase by more double, among which the hypertrophy of fibroblasts and increase in the number are the main reasons for thickening, and collagen fibers also increase. When stimulated by hypoxia, fibroblasts undergo phenotypic changes, then proliferate, differentiate, and upregulate the expression of contractile protein and ECM protein, while simultaneously releasing PASMCs cytokines that affect media. The fibroproliferative changes of vascular adventitia are related to the decreased lumen and decreased responsiveness of the vessel wall to vasodilation. Local hypoxia can also lead to the upregulation of carbonic anhydrase activity in the adventitia, which leads to artery remodeling and inflammatory reactions [103].

### 4.2. PAFs Participate in Adventitia Remodeling

PAFs are the most critical cell component of adventitia, which play a vital role in regulating vascular wall function. When blood vessels are stimulated, adventitial PAFs are activated and undergo phenotypic changes, releasing factors that directly affect the phenotype and growth of PASMCs in media, and recruiting of inflammatory factors and hematopoietic progenitor cells [104].

#### 4.2.1. PAFs Proliferation of Adventitia Remodeling

Hypoxia activates some growth factors; stimulates protein kinase C, the mitogen-activated protein kinase family and phosphatidylinositol 3-kinase (PI3K); and regulates PAFs proliferation [105]. The early proliferation of PAFs caused by acute hypoxia exposure relies on p38 mitogen-activated protein kinase (p38 MAPK), and HIF-1α is also a key transcription factor in the biochemical reaction of hypoxia [106,107]. HIF-1α is the downstream effector of p38 MAPK, and p38 MAPK may also contribute to the stability of HIF-1α. HIF-1α is directly related to p38 MAPK phosphorylation, which is crucial for PAFs proliferation under hypoxia. Furthermore, the downregulation of miR-29 is the basis of TGF-β-mediated PAFs, while HIF-1α and Smad3 can jointly inhibit the expression of miR-29 in organ fibrosis [108]. Further research showed that the TGF-1 receptor blocker could inhibit the proliferation of fibroblasts induced by hypoxia, the expression of TGF-β1, MMP-1, α-SMA and NF-κB, and the increase of collagen fibers in adventitia induced by hypoxia [109].

#### 4.2.2. PAFs Muscularization of Adventitia Remodeling

Hypoxia can induce the proliferation of adventitial PAFs and make them change to myofibroblasts. Myofibroblasts migrate from adventitia to intima or media, resulting in the thickening of the blood vessel wall, as the primary source of collagen, fibronectin, tendon protein and elastin. It is considered a critical participant in pulmonary vascular remodeling [110]. Hypoxia can make the expression of HIF-1α, vascular endothelial growth factor-A (VEGF-A) and matrix metalloproteinases (MMPs) in myofibroblasts upregulated, which promotes the proliferation and migration of myofibroblasts into the intima. 15-Hydroxyeicosatetraenoic acid (15-HETE) is an essential regulatory substance, and HETE inhibitor could reverse the extracellular matrix deposition and 15-lipoxygenase (15-LO) expression [111]. The expressions of myofibroblast markers α-SMA, type I collagen and fibronectin increased in fibroblasts cultured in hypoxia in vitro regulated by 15-LO/15-HETE signaling [112]. 15-HETE can upregulate the expressions of α-SMA and FGF-2, which are also vital participants in cell proliferation and differentiation. It is suggested that hypoxia is involved in regulating the transformation of adventitial fibroblasts into myofibroblasts mainly through the 15-HETE/FGF2/TGF-β1/α-SMA signaling pathway [102].

### 4.3. Vasa Vasorum in Adventitia Remodeling

In arterioles, the intima and media are nourished by the lumen side of the artery wall, while the vasa vasorum nourishes the adventitia. Vasa vasorum originates from the arteries or the arteriovenous nearby [113]. Under hypoxic conditions and early intimal injury, vasa vasorum of the adventitia will generate new blood vessels, which will cause the intima to thicken and penetrate the media and intima. Hypoxia caused the upregulation of HIF-1α/2α expression in adventitia and increased VEGF-A, VEGF-C and FGF-2, promoting angiogenesis. Fibroblasts are the critical regulatory point of adventitia neovascularization. ET-1 is released from fibroblasts activated by hypoxia and vasa vasorum endothelial cells, participating in the regulation of PH with the synergistic effect of VEGF. Furthermore, vasa vasorum in the adventitia of the pulmonary artery with long-term hypoxic expansion secrete ATP, as an essential angiogenic factor. PI3K, Rho and ROCK are involved in ATP-mediated vasa vasorum angiogenesis and assume an important regulatory role in DNA synthesis, migration and tubular structure formation [114]. Adventitia plays an essential role in the development of these diseases. Defining the molecular mechanism of adventitia’s role in hypoxic vascular remodeling and finding the critical action nodes may provide a new target for treating PH.

## 5. ECM Remodeling in Pulmonary Vascular Remodeling

As an extensive molecular network of the cellular surroundings, the ECM provides for the proper functioning of the structure and function of the vessel wall and plays a crucial role in intercellular and intracellular communication [115]. The ECM is a relatively balanced, dynamic environment composed mainly of collagens and many other proteins, including mainly basement membrane (BM), elastin, laminin, collagen IV, fibronectin, tenascin C, proteoglycan, etc. [116,117,118,119]. In addition to the function of ECM components in maintaining tissue structure [120], studies have shown that reduced pulmonary arterial compliance and increased pulmonary artery ECM remodeling play a regulatory role in the development of PH (Figure 5). Although ECM remodeling is still controversial, it does not prevent us from exploring it further as a potential therapeutic target.

### 5.1. The Balance between Proteolytic Enzymes in ECM Remodeling

The composition of ECM is regulated by the balance between proteolytic enzymes, including MMPs, metalloproteases (MPs), serine elastases (SE), lysyl oxidases (LOXs), and their endogenous inhibitors tissue inhibitors of metalloproteinase (TIMPs). Dynamic imbalance of protein hydrolases and endogenous inhibitory cytokines promotes worsened collagen deposition and increased elastin breakdown in the pulmonary artery during PH [121]. The exact mechanisms responsible for these imbalances are not yet known with certainty, but several potential mechanisms are becoming accepted. Exacerbation of flow, shear stress and inflammation as triggering events lead to impaired PAECs, weakening barrier function and increasing permeability, resulting in overproduction of serine elastase content by PASMCs [122]. Serine elastase subsequently leads to ECM degradation and activation of growth factors such as FGF and TGF-β, which subsequently upregulates the deposition of fibronectin, collagen, tenascin and elastin by stimulating PASMCs and PAFs. Furthermore, catabolism of ECM and related growth factors also promotes upregulation of biosynthesis of MMPs, accompanied by an imbalance in protein hydrolases and their inhibitors that can be triggered by inflammation, ultimately leading to remodeling of the ECM [123].

### 5.2. EndoMT in ECM Remodeling

ECM remodeling in PH has been attributed to EndoMT [124,125], in which PAECs develop a mesenchymal phenotype with similar characteristics to PASMCs [126]. Along with reduced expression of endothelial characteristic factors such as endothelial cadherin and platelet endothelial adhesion molecules, intercellular interactions gradually weakened, followed by the separation of PAEC from the intima [127]. Subsequently, PAECs migrate into the media accompanied by upregulated expression levels of α-SMA, collagen, vimentin and MMPs, dedifferentiating into myofibroblast-like mesenchymal cells promoting ECM remodeling by exacerbating the degree of collagen deposition and cross-linking.

Multiple stimuli involved in the course of PH initiate the process of EndoMT. Loss of BMPR2 function has been demonstrated to activate EndoMT [128]. Inhibition of BMPR2 properties in PAECs leads to elevation of high-mobility group AT-hook 1 (HMGA1), a protein that promotes the biosynthesis of a transcriptional regulatory molecule called Slug that increases the amount of α-SMA, accompanied by suppression of endothelial cell genes, as well as the platelet endothelial cell adhesion molecule and vascular endothelial cadherin, ultimately exacerbating the mesenchymal phenotype. Elevated expression levels of the transcription factor Twist can inhibit the biological activity of BMPR2 in EndoMT. Furthermore, stimulation of chronic hypoxia initiates EndoMT by increasing the levels of Snail, β-catenin, NF-κB and STAT3, which regulated by HIF and TGF-β signaling [129]. Inflammation and oxidative stress cytokines such as IL-1β, IL-6, IL-10, TNF-α and ROS contribute to EndoMT via activation of TGF-β mediated inflammatory response and oxidative stress damage.

## 6. Other Pathological Alterations Present in PH

Although a range of abnormal pathological responses results in PH, including increased alveolar exudates and granulomas, and infiltration of peribronchial inflammatory cells, they are generally considered to be concomitant symptoms. We highlight the main changes in perivascular inflammation, progenitor/stem cells, and other molecular mechanisms of PH vascular remodeling.

### 6.1. Perivascular Inflammation

It has been well-documented that perivascular inflammation is an essential prominent pathologic feature and driver for pulmonary vascular remodeling in PH [130,131]. Accumulating evidence suggests a functional role of perivascular inflammation in initiating and progressing pulmonary vascular remodeling in PH [132]. High expression of active cellular chemokines and inflammatory factors correlate with clinical outcomes [133]. Multiple immune cell recruitment occurs in the peripulmonary artery, including neutrophils, macrophages, dendritic cells, mast cells, T lymphocytes, and B lymphocytes [134,135,136]. Vascular and parenchymal cells, such as PAECs, PASMCs and PAFs, change their phenotype, resulting in oversensitivity to inflammatory triggers and active secretion of chemokines that promote the outbreak of inflammatory cascade responses [137].

Furthermore, inflammatory factors can also trigger an imbalance in the levels of protein hydrolases and their inhibitors, which exacerbate the remodeling of ECM around the pulmonary vasculature [123]. Inflammatory factor-mediated levels of ROS upregulate the secretion of MMPs from PAECs, PASMCs and PAFs and reduce the secretion of TIMPs, accompanied by the activation and recruitment of macrophages and neutrophils, which further promotes the secretion levels of MMPs and serine elastase. The breakdown products derived from the action of protein hydrolases have pro-inflammatory effects, promoting inflammatory response and forming a positive feedback regulatory mechanism. In summary, multiple studies have recognized that modulating the cross-talk between inflammatory factors and vascular parenchymal cells may serve as a new and effective target for PH immunotherapy.

### 6.2. Progenitor/Stem Cells in Vascular Remodeling during PH

Explaining the pathological process of PH only by the in situ proliferation of resident vascular cells is considered one-sided; attention should also be paid to the invasion of extravascular cells into the vessel wall. As undifferentiated cells that can be transformed into new vascular cells, these vascular progenitors and stem cells have the ability to self-renew and aggregate together under specific stimuli, which exacerbate the pathological process of pulmonary vascular remodeling in PH under the regulation of related signaling pathways [132]. Given previous studies, as two distinct subpopulations of endothelial progenitor cells (EPCs), early-growing EPCs derived from the hematopoietic lineage and late-growing EPCs originated from the endothelial lineage, have been powerfully demonstrated in PH [138]. Studies have measured a simultaneous increase in expression levels of vascular endothelial cells expressing the markers CD133, CD34 and VEGFR-2 in proportion to mPAP [139,140]. In the pulmonary vascular system of PH patients, an upregulation of the EPCs population with CD31, CD34, vWF, c-kit, eNOS and caveolin-1 as markers was similarly observed [141,142]. Pericytes are used as a source of smooth muscle progenitor cells (SPC), which have also been demonstrated during neomuscularization. Moreover, pulmonary endothelium-pericyte interactions can influence the formation of new blood vessels [143,144]. The side population of progenitor cells labeled with ATP Binding Cassette Subfamily G Member 2 (ABCG2) has the potential to differentiate into α-SMA^+^ SMC/myofibroblasts and NG2^+^ pericytes [145]. There is still a need to understand the role that progenitor/stem cells assume in the PH vascular remodeling process.

### 6.3. Ion Channels

Pulmonary vascular remodeling in PH involves pathobiology such as altered pulmonary arterial tone, endothelial dysfunction, and inflammation, which could all depend on K^+^, Ca2^+^, Na^+^ and Cl^−^ ion channel activities [146]. Membrane permeability to cations and anions control resting membrane potential, intracellular ion homeostasis and cell volume. Pulmonary vascular cells from PH are characterized by an important remodeling of protein expression and function of ion channels [147]. K_v_1.5 isoform is a hypoxia-sensitive voltage-gated K^+^ channel in PASMCs, and its overexpression increases K^+^ currents and leads to plasma membrane hyperpolarization [148,149,150]. A decrease in mRNA expressions for K_v_1.1, K_v_1.5, K_v_4.3, Ca^2+^-activated K^+^ and K_2P_ channel are found in patients of PH. Furthermore, the process of PH is also regulated by voltage-gated Ca^2+^ channels and Ca^2+^-activated Cl^−^ channels, which lead to pulmonary vascular remodeling by promoting membrane depolarization, vasomotor tone and cell excitability [151,152,153]. It has also been shown that the Na^+^ voltage-gated channel subunit 1B (SCN1B) is upregulated in patients with PH, which may contribute to abnormal vasoconstriction and excessive remodeling, and it could be carefully considered as a relevant novel therapeutic target [154].

## 7. Conclusions

Understanding how all these processes integrate to promote the aberrant proliferative remodeling characteristic of PH will be critical to developing successful antiremodeling therapies. The search for novel therapeutic targets is necessary because currently commonly used treatments associated with adverse side effects for PH are imperfect. Although several promising potential therapies have emerged from studies in non-human models, such as dichloroacetate, inhibitors of BMPR2, ROCK and HIF, their therapeutic effects on patients still need to be validated by large-scale clinical trials. Other questions remain, particularly how best to increase the success rate of translating preclinical discoveries to the bedside, which has been a significant challenge in PH. Further investigation is needed to explore the pathogenesis and regulatory pathways of vascular remodeling and prepare the theoretical basis for developing new drugs to target and reverse the remodeling process in PH.

## Figures and Tables

**Figure 1 jpm-13-00366-f001:**
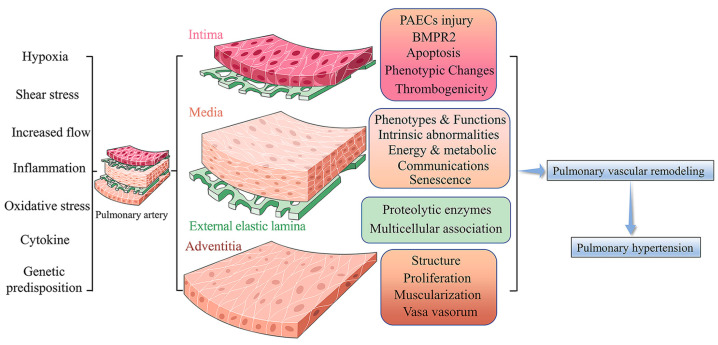
Pulmonary Vascular Remodeling in PH Pathogenesis. Under a variety of stimuli, such as hypoxia, inflammation, and oxidative stress, the pulmonary artery vasculature undergoes pathological changes in function and structure, leading to pulmonary vascular remodeling and ultimately promoting PH.

**Figure 2 jpm-13-00366-f002:**
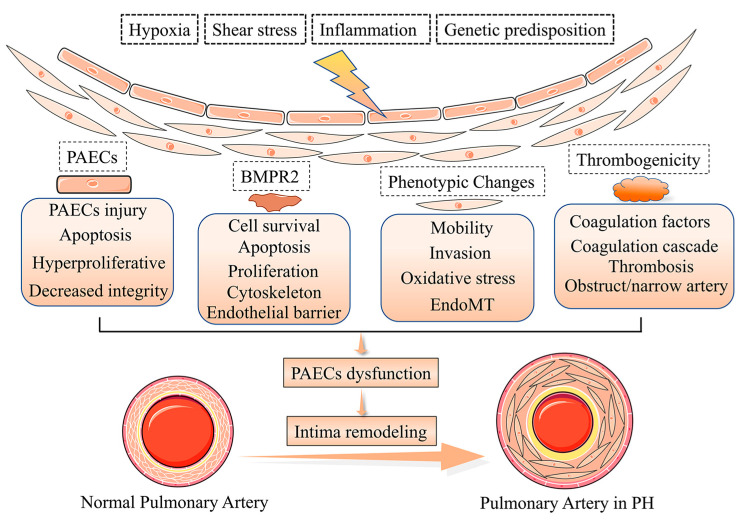
Pathogenesis of pulmonary vascular intima remodeling in PH. In response to hypoxia, shear stress, inflammation, and genetic predisposition, the function and phenotype of PACEs are altered, resulting in intima remodeling in PH.

**Figure 3 jpm-13-00366-f003:**
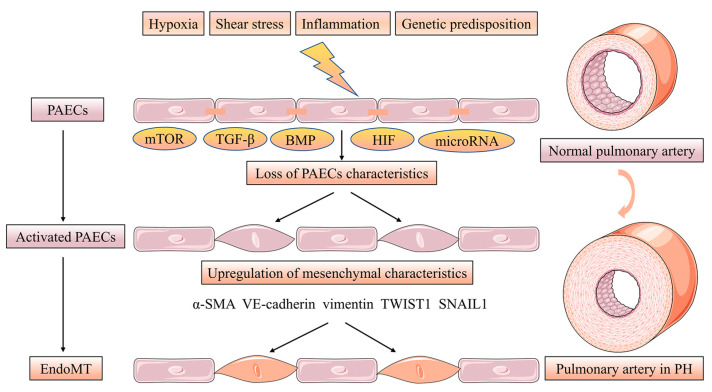
EndoMT of PACEs in pulmonary vascular intima remodeling. EndoMT is a phenotypic change in which PAECs manifesting a mesenchymal-like phenotype with concomitant endothelial cell characteristics loss while upregulating the level of mesenchymal markers. Furthermore, PAECs adopt highly migratory and invasive cell phenotype characteristics with losing cell-cell contact. The EndoMT process is regulated by the mTOR, TGF-β, BMP and HIF signaling pathway. microRNA, such as miR-27a, miR-124 and miR-181b can be implicated in EndoMT in PH.

**Figure 4 jpm-13-00366-f004:**
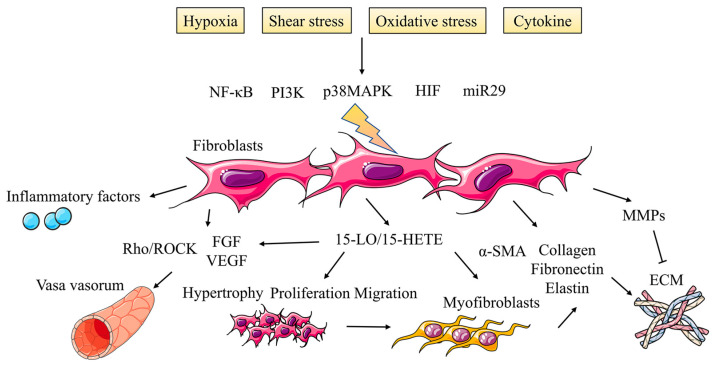
Activated fibroblasts promote adventitia remodeling in PH. During the period of vascular remodeling induced by PH, PAFs are highly activated and undergo phenotypic transformation characterized by hyperproliferation, migration and inflammatory activities under the regulation of multiple signaling pathways, which also affects vasa vasorum and ECM in adventitia.

**Figure 5 jpm-13-00366-f005:**
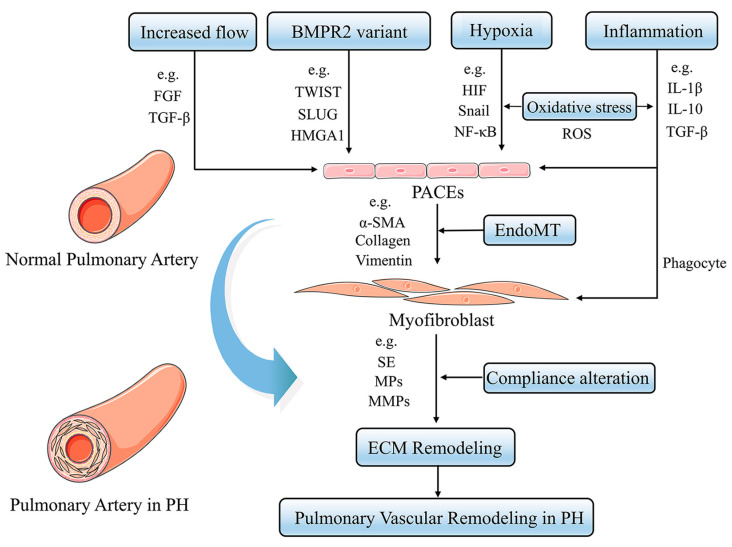
Pathogenesis of pulmonary vascular ECM remodeling in PH. Increased blood flow, hypoxia, inflammation and BMPR2 signaling lead to EndoMT, in which endothelial cells acquire increased expression of the mesenchymal phenotype. These factors then affect the function and structure of the ECM, including MMPs, collagen, vimentin, etc., resulting in ECM remodeling.

## Data Availability

Not applicable.

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
