# Peer review of "Pulmonary Vascular Remodeling in Pulmonary Hypertension"

_jpm, 2023, doi:10.3390/jpm13020366_

Round 1

Reviewer 1 Report

This review is not adding so much to the existing body of knowledge on the topic of Pulmonary Arterial HypertensionThe quality can be improved by adding some personal/original interpretation of the actual body of knowledge and some future perspectives. Some parts of the text are quite difficult to understand, due to poor English language. Improving the language by a native English speaker might make it reasonably acceptable.

Reviewer 2 Report

The review "Pulmonary vascular remodeling in pulmonary hypertension" " by Z. Jia et al. gives an overview of vascular remodeling in pulmonary hypertension. The manuscript is organized in seven sections. The first section corresponds to a general reminder of pulmonary hypertension. The second, third and fourth sections concern the intima, media and adventitia remodeling, respectively. The fifth section focuses on the extracellular matrix remodeling. The sixth section concerns other pathological alterations occurring during pulmonary hypertension. The last section is the conclusion. In my view, this review is generally well organized, well written and is a good synthesized overview. However, there are some points that need to be addressed to improve its quality.   Content:   1/ While ion channels play an important role in vascular remodeling, their involvement is only very furtively mentioned twice in the manuscript (lines 206 and 215). A paragraph detailing their role should be added in each of sections 2-3 (i.e. intima, media and adventitia remodeling).   2/ Similarly, the transcription factor NFAT is also involved in vascular remodeling, but only NF-κB is mentioned (lines 208, 304 and 392).

Language:   3/ The authors should carefully reread the manuscript: throughout the text, there are some grammatical and typographical mistakes to correct. For example: - lines 17 and 45: "layer" instead of "membranous" - fig.1: "external elastic lamina" instead of "extracellular matrix" - lines 116-120, 144-146, 152-154, and 341-344: the sentences are difficult to understand - line 125: give the abbreviation of PPAR that appears only on line 207 - line 132: supress "are" and "s" in the sentence "mTor, Nur77 and GDF11 ARE also actS as inhibitors…" - lines 134 and 216: "Ca2+" instead of "Ca2+" - lines 206: "K+" instead of "K+" - line 308: replace “intima” by “media” - lines 391: "stimulation" instead of "Furthermore, Stimulation…"

Reviewer 3 Report

In this minireview, the authors reviewed pathological changes and highlights some pathogenetic mechanisms involved in the remodeling process during the development of pulmonary hypertension. The manuscript is well organized and some of the information is well-presented in figures. Before publishing, the authors need to address the following issues.

1.       The authors used the term pulmonary hypertension which includes 5 major groups. It is difficult to discuss all the groups of PH in this review, thus the authors should just use pulmonary arterial hypertension (PAH) in this review.

2.       When describing some novel target genes in PH, the authors need to cite the original paper of the findings. For example, the role of FoxO1 in PH was first described extensively in a Nature Medicine paper (PMID: 25344740) in 2014.

3.       The authors did not provide clear evidence of EC proliferation in regulation of PH. Actually, this is still a controversial topic in the field, please add a brief discussion on this.

4.       Based on my reading of the literature, I recommend the authors include these recent important findings (or even a brief discussion) in their reference. Pericytes in PH: PMID: 30586764; Metabolic shift in PASMC: PMID: 30242159, PMID: 30817168; other PH related gene mutations (besides the BMPR pathway): PMID: 36205124, PMID: 32079640 etc...

5.       Please double check the English writing in the manuscript.
